# RNA-Seq Reveals the Roles of Long Non-Coding RNAs (lncRNAs) in Cashmere Fiber Production Performance of Cashmere Goats in China

**DOI:** 10.3390/genes14020384

**Published:** 2023-02-01

**Authors:** Xinmiao Wu, Yuanhua Gu, Shiqiang Li, Shiwei Guo, Jiqing Wang, Yuzhu Luo, Jiang Hu, Xiu Liu, Shaobin Li, Zhiyun Hao, Mingna Li, Bingang Shi

**Affiliations:** Gansu Key Laboratory of Herbivorous Animal Biotechnology, College of Animal Science and Technology, Gansu Agricultural University, Lanzhou 730070, China

**Keywords:** long non-coding RNA, cashmere fiber, cashmere goats, skin, RNA-seq

## Abstract

Long non-coding RNAs (lncRNAs) are a kind of non-coding RNA being >200 nucleotides in length, and they are found to participate in hair follicle growth and development and wool fiber traits regulation. However, there are limited studies reporting the role of lncRNAs in cashmere fiber production in cashmere goats. In this study, Liaoning cashmere (LC) goats (*n* = 6) and Ziwuling black (ZB) goats (*n* = 6) with remarkable divergences in cashmere yield, cashmere fiber diameter, and cashmere color were selected for the construction of expression profiles of lncRNAs in skin tissue using RNA sequencing (RNA-seq). According to our previous report about the expression profiles of mRNAs originated from the same skin tissue as those used in the study, the *cis* and *trans* target genes of differentially expressed lncRNAs between the two caprine breeds were screened, resulting in a lncRNA–mRNA network. A total of 129 lncRNAs were differentially expressed in caprine skin tissue samples between LC goats and ZB goats. The presence of 2 *cis* target genes and 48 *trans* target genes for the differentially expressed lncRNAs resulted in 2 lncRNA-*cis* target gene pairs and 93 lncRNA-*trans* target gene pairs. The target genes concentrated on signaling pathways that were related to fiber follicle development, cashmere fiber diameter, and cashmere fiber color, including PPAR signaling pathway, metabolic pathways, fatty acid metabolism, fatty acid biosynthesis, tyrosine metabolism, and melanogenesis. A lncRNA–mRNA network revealed 22 lncRNA-*trans* target gene pairs for seven differentially expressed lncRNAs selected, of which 13 *trans* target genes contributed to regulation of cashmere fiber diameter, while nine *trans* target genes were responsible for cashmere fiber color. This study brings a clear explanation about the influences of lncRNAs over cashmere fiber traits in cashmere goats.

## 1. Introduction

Cashmere fiber is a kind of precious and special animal fiber, with features of being slender, softer, and warmer. The fiber is also called “soft gold” [1]. Many studies have shown that a great number of functional genes and non-coding RNAs are responsible for cashmere fiber quality and quantity regulation in cashmere goats [2,3]. Therefore, it is possible to improve fiber traits by regulating the expression of non-coding RNAs. 

Long non-coding RNAs (lncRNAs) are a kind of non-coding RNAs with a length of >200 nucleotides in length. The expression of lncRNAs is lower than their corresponding protein coding genes and they also have higher tissue-specificity. Meanwhile, lncRNAs are more evolutionarily conserved [4]. The lncRNAs plays their biological roles in a variety of biological phenomena and life processes through multiple mechanisms. For example, some lncRNAs change the transcription of functional genes in *trans* or *cis* [5]. Additionally, some lncRNAs reduce the inhibited effect on mRNAs by miRNAs by playing the role of miRNA sponges. Moreover, lncRNA can regulate the structure and function of chromatin through interaction with DNA, RNA and protein, regulate trace and histone modifications through transcriptional interference and chromatin-mediated inhibition, and participate in the formation and regulation of organelles and nuclear solidified substances [6].

The lncRNAs have been affirmed to contribute to hair follicle development and fiber traits regulation in sheep and cashmere goats. For example, TCONS_00279168 targeted *ATP1B4* and *FGF12* that were associated with ovine wool follicle development [7]. The lncRNA XLOC_008679 regulated cashmere fiber diameter of cashmere goats by targeting *KRT35* [8]. However, up to now, research into lncRNAs of skin tissues in cashmere goats has mainly been focused on the expression profiles constructed at different development stages or breeds with different cashmere fiber traits using RNA sequencing (RNA-seq). For example, there were 173 differentially expressed lncRNAs screened in cashmere goat skins between catagen and anagen phases of fiber follicle growth, and their target genes were related to fiber follicle regression [9]. A total of 80 differentially expressed lncRNAs were identified in caprine skin tissue samples between the fine-type and coarse-type Tibetan cashmere goats, and their target genes were responsible for the fiber follicle growth [10]. Additionally, 57 differentially expressed lncRNAs were found between white, black, and brown skin of cashmere goats, and they were involved mainly in melanogenesis; PI3K-Akt and MAPK signaling pathways that were associated with cashmere fiber color and pigmentation [11]. However, there have been no comparisons about skin tissue expression profiles of lncRNAs between different caprine breeds.

Liaoning cashmere (LC) goats and Ziwuling black (ZB) goats are the main cashmere goat breeds in China and play economic importance roles for goat farmers for which they are raised. The two breeds have distinct different cashmere fiber traits. LC goats produce pure white cashmere fiber with higher fiber yield and fiber diameter. In contrast, ZB goats produce purple cashmere fiber with lower fiber yield and fiber diameter [12]. Specifically, the average cashmere fiber yield and cashmere fiber diameter of LC goat are 1300 g and 15.5 μm, respectively, while the average cashmere yield and cashmere fiber diameter of ZB goats are 310 g and 14.1 μm, respectively [13]. However, the molecular mechanism that regulates the differences in cashmere fiber traits between LC goats and ZB goats remains unclear. Accordingly in this study, the lncRNA expression profile of skin tissue collected from LC goats and ZB goats were investigated by RNA-seq analysis. The functional enrichment was also analyzed for the target genes of differentially expressed lncRNAs. The results will provide possible function of lncRNAs in regulation of cashmere fiber traits in cashmere goats. 

## 2. Materials and Methods

### 2.1. Sample Collection

Based on the permission of Animal Experiment Ethics Committee of Gansu Agricultural University (Approval number GSAU-ETH-AST-2021-028), experimental animals were operated and treated.

Six healthy, three-year-old male LC goats and six healthy, three-year-old male ZB goats were chosen for investigation. All the goats were raised at the same feeding and management levels at Yongfeng Cashmere Goat Breeding Company in Huan county (Qingyang, China). The weight and mean diameter of cashmere fibers of the LC goats investigated in the study were 1539 ± 26.6 g and 15.9 ± 0.07 μm, while the traits of the ZB goats were 395 ± 17.5 g and 13.9 ± 0.03 μm, respectively. 

In August, when these goats were in the anagen phase of fiber growth, skin tissue samples were gathered from the right mid-side of body from each goat. Briefly, after shearing and disinfecting with 75% alcohol, these goats were locally anesthetized with 2% lignocaine. A skin sample of 3 cm × 2 cm in size was gathered. The samples were rinsed using normal saline and RNase water. After absorbing the water, the edge of skin tissue was trimmed, and the samples were cut into a piece of 1 cm × 2 cm. The samples were promptly frozen in liquid nitrogen. 

### 2.2. RNA Isolation and RNA-Seq

Total RNA from the skin samples was extracted and purified according to the method of Bao et al. [14]. The Agilent 2100 Bioanalyzer (Agilent, Santa Clara, CA, USA) was used to select high-quality RNA samples with RNA Integrity Number value > 7. The removing of Ribosome RNA (rRNA) was carried out from the high-quality RNA using a Ribo-Zero Gold rRNA Removal Kit (Illumina, San Diego, CA, USA). The remaining RNA was reverse transcribed into complementary DNA (cDNA) using a NEBNext Ultra RNA Library Prep Kit (New England Biolabs, Ipswich, MA, USA). The cDNA libraries constructed were paired-end sequenced using an Illumina Novaseq 6000 sequencer (Illumina, San Diego, CA, USA) at the Gene Denovo Biotechnology Co., Ltd. (Guangzhou, China).

### 2.3. RNA-Seq Data Processing and Screening of LncRNAs in Caprine Skin Tissue

To get high- quality clean reads, raw reads were further filtered by removing reads, including adapters, reads including more than 10% of unknown nucleotides, and low-quality reads, whose quality score was less than Q20 using fastp V0.18.0 [15]. The clean reads were aligned to Caprine Genome Assembly ARS1.2 using HISAT2 V2.1.0 [16], and known lncRNAs were annotated based on annotation information of existing lncRNAs on the genome assembly. 

The Stringtie V1.3.4 was used to reconstruct transcripts, and all reconstructed transcripts were compared to Caprine Genome Assembly ARS1.2. The results were classed into twelve parts using Cuffcompare. Transcripts with one of class codes were selected, including u (intergenic transcripts), i (transcripts completely within intron), j (multiple exons with at least one joint match), x (exonic overlap with reference on the opposite strand), c (intron compatible), e (single exon transfrag partially covering an intron, possible pre-mRNA fragment), and o (other same strand overlap with reference exons). The transcripts both with a length of longer than 200 bp and with ≥2 exons were further selected. Two kinds of software, Coding-Non-Coding Index V2 [17] and Coding Potential Calculator V0.9-r2 (http://cpc.cbi.pku.edu.cn/, accessed on 27 June 2020), were operated for predicting the scores of protein-coding potentials for the transcripts selected. The intersected data from two kinds of software were chosen as novel lncRNAs. The differences in some data between lncRNAs and mRNA originated from the same skin samples as those used in this study were compared including the number of exon number, the length of transcript and open reading frame, and expression levels.

### 2.4. Screening of Differentially Expressed LncRNAs and Validation of the RNA-Seq Results

Each lncRNA screened in expression was calculated using the Fragments Per Kilobase of transcript per Million reads mapped (FPKM) using StringTie V1.3.4. The differentially expressed lncRNAs were screened in skin tissue between LC goats and ZB goats using the DESeq V2.0 [18]. The lncRNAs with both |fold change)| > 2 and *p*-value < 0.05 were considered to be up-regulated or down-regulated lncRNAs in LC goats. The lncRNAs with the minimum *p*-value were considered as the most significant differentially expressed lncRNAs.

A random selection of nine differentially expressed lncRNAs were used to verify the reliability of the RNA-seq data using reverse transcriptase-quantitative polymerase chain reaction (RT-qPCR) analysis. The RNA samples used initially for RNA-seq were used to produce cDNA using SuperScriptTM II reverse transcriptase (Invitrogen, Waltham, MA, USA). The RT-qPCR was carried out using 2 × ChamQ SYBR qPCR Master system (Vazyme, Nanjing, China), and three technical repetitions were set. *GAPDH* was selected as an internal reference [10]. A 2^−ΔΔCt^ was calculated for normalizing the lncRNAs in expression. The primer information of RT-qPCR is shown in Table 1.

### 2.5. Functional Enrichment Analysis of the Target Genes of Differentially Expressed LncRNAs and a Network Construction of LncRNA-mRNA

To accurately predict the target genes of differentially expressed lncRNAs, the differentially expressed genes were screened from the same skin tissues as those used in this study using the thresholds of |fold change| > 2 and *p*-value < 0.05. The differentially expressed genes located within 100 kb of differentially expressed lncRNAs were selected as *cis* target genes. Additionally, Pearson correlation coefficient in expression of differentially expressed lncRNAs with differentially expressed genes were calculated, and the genes with |*r*| > 0.95 and *p*-value < 0.05 were selected as *trans* target genes of the lncRNAs.

For the *cis* and *trans* target genes of differentially expressed lncRNAs screened, their Gene Ontology (GO) and Kyoto Encyclopedia of Genes and Genomes (KEGG) functional enrichments were analyzed. The significant GO terms and KEGG pathways with *p* < 0.05 were screened based on a hypergeometric test.

Of the target genes predicted, the genes associated with cashmere fiber traits and their corresponding lncRNAs were chosen to build a lncRNA-mRNA figure using Cytoscape V3.5.1.

### 2.6. Prediction of Binding miRNAs of Seven LncRNAs Selected

The binding miRNAs of seven lncRNAs selected were analyzed using miRBase V21 [19] and miRPara V6.3 [20], and the results from the two types of software were compared.

## 3. Results

### 3.1. Overview of RNA-Seq Data

The average of 84,273,195 and 93,249,211 raw reads were obtained from skin tissue samples of LC goats and ZB goats, respectively. Our raw reads have been submitted to GenBank with accession numbers SRR19879981-SRR19879992. After quality control and filtering of the raw reads, 84,025,004 and 92,946,813 clean reads were obtained, of which 93.46% and 92.41% of clean reads were uniquely aligned to Caprine Genome Assembly ARS1.2, respectively. These indicate that RNA-seq data obtained was highly reliable, ensuring the accuracy of subsequent analysis.

### 3.2. Screening and Expression of LncRNAs

A total of 3934 lncRNAs were obtained in caprine skin tissue, including 2676 known lncRNAs and 1258 novel lncRNAs (Figure 1A; Appendix A). A total of 2434 lncRNAs were co-expressed in both LC and ZB goats, while 714 and 786 lncRNAs were specifically expressed in LC and ZB goats, respectively. 

Based on FPKM value of each lncRNA, the expression distribution of lncRNAs in different samples is shown in Figure 1B. It can be inferred that the trend of lncRNAs in expression is basically similar and stable for 12 samples as they had the same peak. However, the overall expression level of lncRNAs is low (Figure 1B,C).

### 3.3. Characteristic of LncRNAs

Of the five types classified in view of the position of lncRNAs in Caprine Genome Assembly ARS1.2 relative to structural genes, intergenic lncRNA was the most common type, accounting for 56.2%, followed by antisense lncRNAs (12.8%), sense lncRNAs (10.5%), and bidirectional lncRNAs (10.4%). Intronic lncRNAs (2.4%) were the least common type. Additionally, 7.7% of lncRNAs were considered to be other type of lncRNAs (Figure 2A). 

We compared the differences in exon number, the length of transcript and open reading frame, and expression levels between lncRNA and mRNA obtained from the same skin tissues as those used in this investigation. The results showed that known and novel lncRNAs had 3.7 and 2.4 exons on average, respectively, and the number was far less than mRNAs with average exons of 12.6 (Figure 2B). The average length of known lncRNAs (1433 bp) was lower than novel lncRNAs and mRNAs that they had an average length of 5478 bp and 3477 bp, respectively (Figure 2C). The average length of open reading frame and abundance of lncRNAs were also lower than those of mRNAs (Figure 2D,E). The coding potential score of mRNAs was the largest, followed by novel lncRNAs and known lncRNAs (Figure 2F).

### 3.4. Differential Expression Analysis and Verification of LncRNAs

A total of 129 lncRNAs were differentially expressed in skin samples between LC goats and ZB goats, including 38 up-regulated lncRNAs and 91 down-regulated lncRNAs in LC goats (Figure 3; Appendix A). Of these up-regulated lncRNAs in LC goats, the most prominent was MSTRG.9141.2, with a 4.27-fold increase, followed by MSTRG.5223.7, MSTRG.8274.1, MSTRG.2390.1, and XR_001917125.1. However, of these down-regulated lncRNAs in LC goats, MSTRG.17213.1 was the most significant down-regulated lncRNA, with a 135.55-fold reduction, followed by XR_001296368.2, MSTRG.17213.2, MSTRG.18609.1, and MSTRG.14014.1. 

A total of nine lncRNAs selected were used to verify the reliability of RNA-seq data. The data produced from the RT-qPCR showed that the tendency of the lncRNAs in expression was in consistency with that derived from the RNA-seq analysis (Figure 4), thus demonstrating the repeatability and reliability of RNA-seq analysis.

### 3.5. Function Enrichment Analysis of the Target Genes of Differentially Expressed LncRNAs

There were two *cis* target genes and 48 *trans* target genes obtained for two and 25 differentially expressed lncRNAs, respectively. These resulted in two lncRNA-*cis* target gene pairs and 93 lncRNA-*trans* target gene pairs (Appendix A).

The *cis* and *trans* target genes significantly concentrated on 166 biological process (GO-BP) terms, 15 cellular components (GO-CC) terms, and 29 molecular function (GO-MF) terms (Appendix A). Interestingly, some terms were directly associated with cashmere fiber synthesis, including single-organism metabolic process (*p* = 1.872 × 10^−4^), sulfur compound biosynthetic process (*p* = 0.049), and lipid metabolic process (*p* = 8.689 × 10^−4^), while some GO terms were related to the regulation of cashmere fiber diameter, including fatty acid metabolic process (*p* = 5.052 × 10^−7^), thioester metabolic process (*p* = 1.506 × 10^−4^), fatty acid biosynthetic process (*p* = 0.005), and metabolic process (*p* = 0.007). Some terms associated with pigmentation were also found, including pigmentation (*p* = 4.911 × 10^−4^), melanosome localization (*p* = 7.477 × 10^−4^), and pigment granule localization (*p* = 0.001) (Figure 5A). 

The *cis* and *trans* target genes obtained were significantly concentrated on 15 KEGG pathways (Appendix A). Some KEGG pathways associated with fiber diameter were found, including metabolic pathways (*p* = 1.798 × 10^−4^), lipoic acid metabolism (*p* = 2.968 × 10^−4^), fatty acid degradation (*p* = 5.529 × 10^−4^), fatty acid metabolism (*p* = 6.222 × 10^−4^), fatty acid biosynthesis (*p* = 0.001), and adipocytokine signaling pathway (*p* = 0.026). Notably, the PPAR signaling pathway was associated with fiber follicle development (*p* = 2.170 × 10^−4^), while tyrosine metabolism (*p* = 4.019 × 10^−4^) and melanogenesis (*p* = 0.004) were related to pigmentation (Figure 5B). 

### 3.6. A LncRNA and mRNA Network

Of the 95 lncRNA-mRNA pairs described above, according to the functions of the mRNAs in regulation of cashmere fiber traits, a total of 22 lncRNA-mRNA pairs were selected to build a lncRNA-mRNA network, including nine pairs associated with pigmentation and 13 pairs related to cashmere fiber diameter (Figure 6). For example, MSTRG.17213.1 would regulate the expression levels of tyrosinase related protein 1 (*TYRP1*), tyrosinase (*TYR*), ceruloplasmin (*CP*), solute carrier family 45 member 2 (*SLC45A2*), and solute carrier family 24 member 5 (*SLC24A5*), and MSTRG.9141.2 would regulate the expression level of agouti signaling protein (*ASIP*). The two lncRNAs participated in regulation of cashmere color by targeting the mRNAs. However, MSTRG.13442.1 and MSTRG.9213.4 influenced the diameter of cashmere fiber by changing the expression of homeobox A7 (*HOXA7*).

### 3.7. The Association Analysis of LncRNA with miRNA

Of the differentially expressed lncRNAs identified in the study, only one lncRNA was predicted to target two miRNAs, eventually producing two lncRNA-miRNA pairs, including MSTRG.4883.1-miR-143 and MSTRG.4883.1-miR-145. They were related to pigmentation and fiber follicle development.

## 4. Discussion

Cashmere fiber is the raw material of all kinds of textiles, and its production can bring huge economic benefits to goat industry. The characteristics of cashmere fiber can be regulated by lncRNAs, but there are few studies on the effect of lncRNAs on cashmere fiber traits. In this study, a total of 3934 lncRNAs were found in the skin tissue of LC goats and ZB goats. The amount of lncRNAs identified in the investigation was more than those found in skin tissue of Tibetan cashmere goats, with 2059 lncRNAs being found [10], but less than 3975 lncRNAs reported in skin tissue of Tibetan carpet sheep [21]. It indicates the breed-specific expression patterns of lncRNAs. The overall expression of lncRNAs found was low, which was consistent with the research of Li et al. [22]. By comparing the characteristics of lncRNAs with mRNAs, it was found that the number of exons, the length of transcript and open reading frame, and abundance of lncRNAs were less than those of mRNAs. This was also consistent with the findings of lncRNAs in skin tissue of Tibetan cashmere goats described by Fu et al. [10]. These results suggest the reliability of our data.

MSTRG.9141.2 was the most up-regulated lncRNA in LC goats. The lncRNA would *trans*-regulate the expression of *ASIP* (Figure 6). The essential roles of *ASIP* in the formation of white wool have been reported previously. For example, *ASIP* can inhibit the formation of eumelanin, thus resulting in a lighter coat color [23]. Xiong et al. [24] detected higher expression of *ASIP* in white goats compared to black goats. Meanwhile, the increased expression of *ASIP* in Japanese quail caused its white plumage [25]. Conversely, *ASIP* in expression of the skin of Youzhou black goats decreased significantly [26]. These results may partly explain why LC goats with higher expression of MSTRG.9141.2 produce white cashmere fiber. 

Some down-regulated lncRNAs in LC goats would mainly regulate the expression of genes related to pigmentation. For example, MSTRG.17213.1 and MSTRG.17213.2 were the most down-regulated lncRNAs in LC goats. MSTRG.17213.1 would *trans*-regulate the expression levels of *TYRP1*, *TYR*, *CP*, *SLC45A2,* and *SLC24A5*, while MSRG.17213.2 would *trans*-regulate the expression of *TYR*, *SLC45A2,* and *SLC24A5* (Figure 6). The increased expression of *TYR* and *TYRP1* were found to lead to pigmentation [27]. Anello et al. [28] found significantly lower expression of *TYR* in the white group when compared to the non-diluted group of llamas. *CP* participates in the metabolism of copper ions, and their amount directly affects the color, quantity, and distribution of fiber pigment particles in cortical cells [29]. *SLC45A2* and *SLC24A5* were related to melanin synthesis [30]. These results explain, again, the divergences in cashmere fiber color between LC goats and ZB goats by differentially expressing MSTRG.17213.1 and MSTRG.17213.2.

Lipid synthesis and metabolism have been confirmed to have crucial effect on hair follicle metabolism and cashmere fiber diameter regulation [31], and they also were related to wool synthesis [32]. Specifically, lanolin is secreted by the sebaceous gland of hair follicle, which can soften wool fiber. The expression of lipid metabolic related genes within ovine skin altered the content of lanolin [2]. Taken together, it was concluded that lipid metabolic related genes may be associated with cashmere fiber diameter. In the study, between some down-regulated lncRNAs in LC goats with some functional genes related to lipid synthesis and metabolism, lncRNAs-*trans* mRNAs pairs were found, including MSTRG.10536.1-apolipoprotein L6 (*APOL6*), MSTRG.10536.1-elongation of very long chain fatty acids protein 7 (*ELOVL7*), MSTRG.10536.1-Long-chain acyl-CoA dehydrogenase (*ACADL*), MSTRG.13442.1-*APOL6*, MSTRG.13442.1-*ACADL*, MSTRG.3149.1-*ACADL*, MSTRG.3149.1-peroxisome proliferator activated receptor γ (*PPARG*), MSTRG.3149.1-*ELOVL7*, MSTRG.9213.4-*PPARG*, MSTRG.9213.4-*ELOVL7*, MSTRG.13442.1-acetyl-CoA carboxylase β (*ACACB*), and MSTRG.9213.4-*ACACB* (Figure 6). APOL6 is a protein of *Bcl-2* homologous 3, which can promote cell differentiation and fat formation [33]. *PPARG* is the main gene involved in cell fat deposition [34]. *ELOVL7* participates in fatty acid prolongation, polyunsaturated fatty acid, saturated fatty acid, and very long-chain fatty acid biosynthetic process [35]. *ACADL* and *ACACB* were related to ester metabolism [36,37]. In this context, MSTRG.10536.1, MSTRG.13442.1, MSTRG.3149.1, and MSTRG.9213.4 may be responsible for differences in cashmere fiber diameter between LC goats and ZB goats by *trans*-regulating the expression of the target genes.

The two down-regulated elements—MSTRG.13442.1 and MSTRG.9213.4 in LC goats—attracted our attention, as they would *trans*-regulate the expression of *HOXA7* (Figure 6). *HOXA7* is one of the most common transcripts detected in human skin [38], and there was a significant positive correlation between the abundance of *HOXA7* and wool fiber diameter in Aohan fine-wool sheep [39]. It could be therefore inferred that the two lncRNAs may be responsible for the difference in cashmere fiber diameter between the two breeds investigated. 

It was notable that some GO terms enriched by the *trans* mRNAs were involved in the regulation of cashmere fiber traits. For example, single-organism metabolic process and sulfur compound metabolic process were related to wool morphogenesis and cashmere synthesis [40]. Some target genes were enriched in various metabolic pathways, including metabolic process, fatty acid metabolic process, cellular lipid metabolic process, and lipid metabolic process. The metabolic pathways have been found to be involved in regulation of wool fiber diameter [41]. According to the position of lipids in the wool fiber, they can be divided into external and internal lipids [42]. Fatty acid is one of components of internal lipids [43]. Therefore, fatty acid transport, fatty acid biosynthetic process, and lipid localization may play roles in regulation of cashmere fiber. Meanwhile, melanin metabolic process, pigment metabolic process, pigmentation, melanosome localization, pigment granule localization, and pigment granule enriched in the study have been found to relate with pigmentation [26].

The *trans* mRNAs of some differentially expressed lncRNAs were enriched in lipoic acid metabolism associated with fiber diameter [41], including acyl-CoA synthetase medium chain family member 2B (*ACSM2B*) and acyl-CoA synthetase medium chain family member 1 (*ACSM1*). Meanwhile, some lipid-related pathways were also enriched by the *trans* mRNAs, including fatty acid degradation, fatty acid metabolism, fatty acid prolongation, and adipocytokine signaling pathways. Approximately 90% of wool or cashmere fibers are protein and the remaining components are lipid [31]. This indicates that lipid metabolism is closely related with cashmere fiber synthesis. As one of the most enriched pathways, PPAR signaling pathway has been found to affect fiber follicle development [41]. The target genes were also enriched in tyrosine metabolism and melanogenesis related to pigmentation [3]. In short, the pathways enriched by the *trans* mRNAs for lncRNAs screened were involved in regulation of cashmere fiber traits of cashmere goats.

LncRNAs can act as miRNA sponges, eventually lessening inhibited influence on mRNAs by miRNAs [44]. In the study, down-regulated MSTRG.4883.1 was estimated to be a sponge of miR-143 and miR-145. The miR-143 has been reported to inhibit the proliferation of dermal papilla cells and delay the cell cycle, and it can influence the activities of hair follicles by targeting *KRT71* in Hu Sheep [45]. The miR-143 also promoted migration and proliferation of melanocyte in mammals, and it is therefore closely associated with hair color [46]. The miR-145 was reported to influence hair follicle development in cashmere goats [47]. This suggests that MSTRG.4883.1 was related to cashmere color and follicle development by sponging miR-143 and miR-145.

## 5. Conclusions

A total of 129 lncRNAs were differentially expressed in the skin tissue samples between LC and ZB goats. The target genes of differentially expressed lncRNAs were associated with cashmere fiber color, cashmere fiber diameter, and cashmere fiber synthesis. This study establishes a solid underpinning for further investigation into the influence of individual lncRNAs in the regulation of cashmere fiber traits in cashmere goats, with an aim of improving cashmere fiber traits by manipulating the expression of the lncRNAs in cashmere goats. 

## Figures and Tables

**Figure 1 genes-14-00384-f001:**
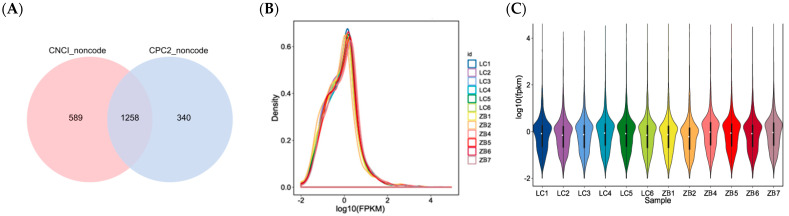
Identification of novel lncRNAs and the expression of the lncRNAs identified. (**A**) Venn diagram describing predicted results from CPC2 and CNCI software. (**B**) The expression abundance of all lncRNAs in 12 libraries. The *X* axis exhibits calculated result of log10 (FPKM), while the *Y* axis shows the density of expressed lncRNAs in samples collected from Liaoning cashmere (LC) goats and Ziwuling black (ZB) goats. (**C**) Comparison of expression levels of lncRNAs in skin samples between LC goats and ZB goats. The *Y* axis indicates calculated result of log10 (FPKM).

**Figure 2 genes-14-00384-f002:**
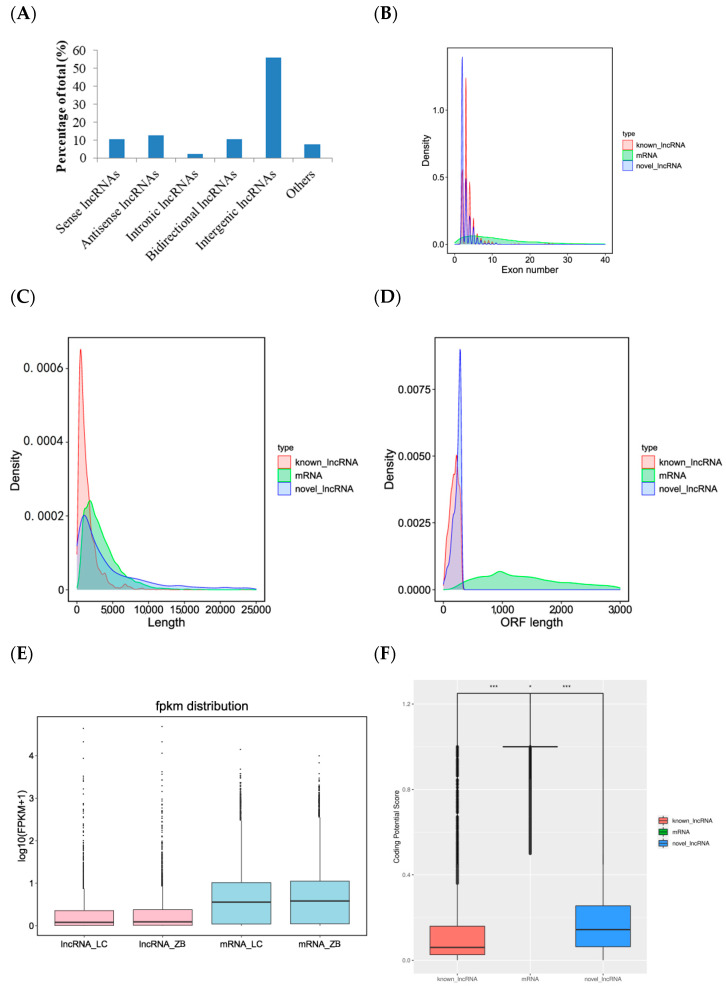
Characteristics of lncRNAs identified in the skin tissue of Liaoning cashmere (LC) and Ziwuling black (ZB) goats. (**A**) Type distribution of lncRNAs identified in skin tissue of cashmere goats. Comparison of exon number (**B**), the length of transcript (**C**), open reading frame (ORF) (**D**), FPKM distribution (**E**), and coding potential score (**F**) between lncRNAs identified in the study and mRNAs originated from the same skin tissue as those used in the investigation. * *p* < 0.05 and *** *p* < 0.001.

**Figure 3 genes-14-00384-f003:**
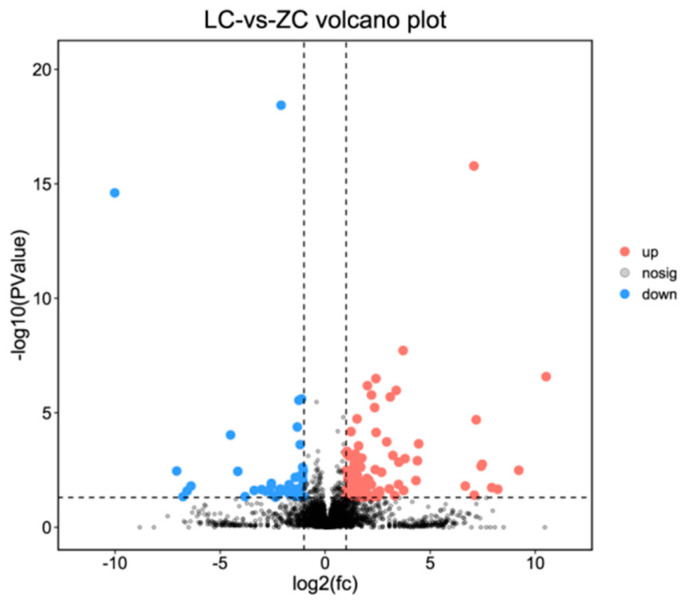
Volcano plot describing the change of lncRNAs in expression between Liaoning cashmere (LC) goats and Ziwuling black (ZB) goats.

**Figure 4 genes-14-00384-f004:**
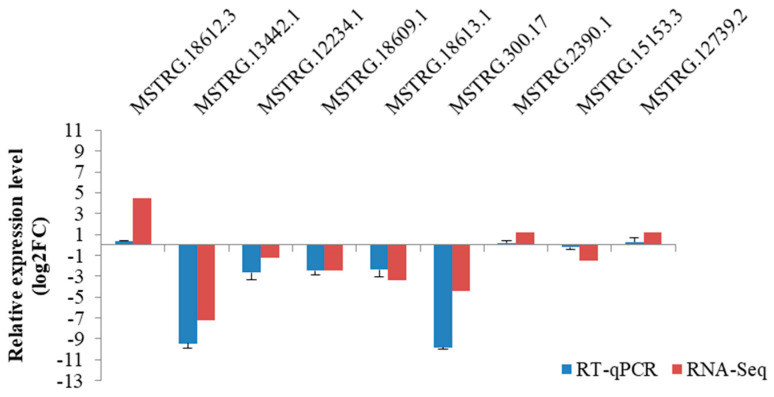
RT–qPCR verification of nine lncRNAs selected in caprine skin tissue identified using RNA–Seq.

**Figure 5 genes-14-00384-f005:**
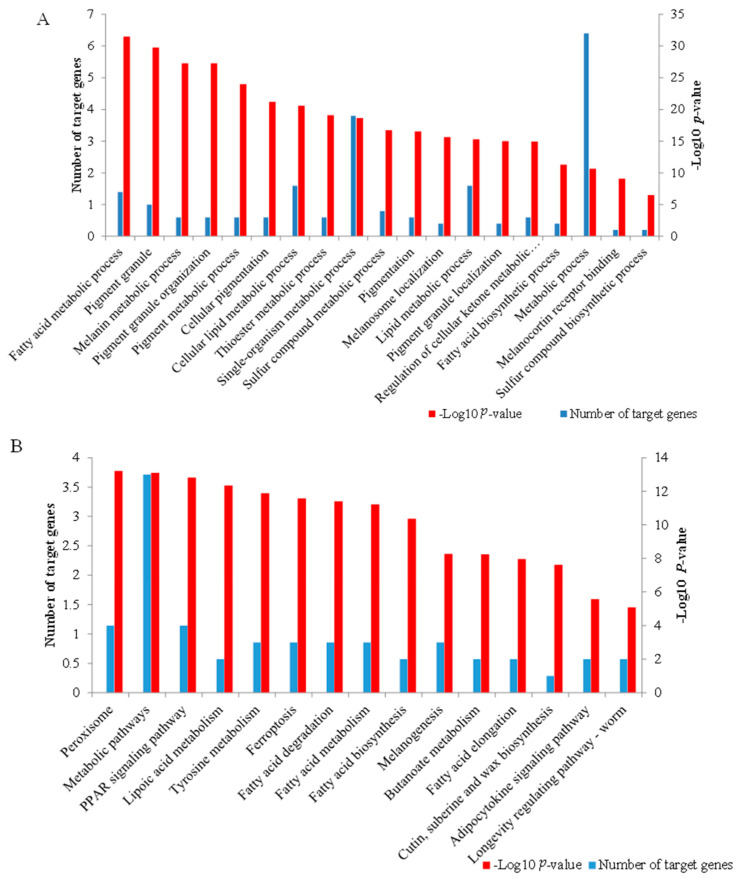
Function enrichment analysis of the target genes of differentially expressed lncRNAs. (**A**) The important GO terms associated with cashmere fiber synthesis, fiber diameter, and pigmentation. (**B**) The 15 significant pathways enriched by the target genes.

**Figure 6 genes-14-00384-f006:**
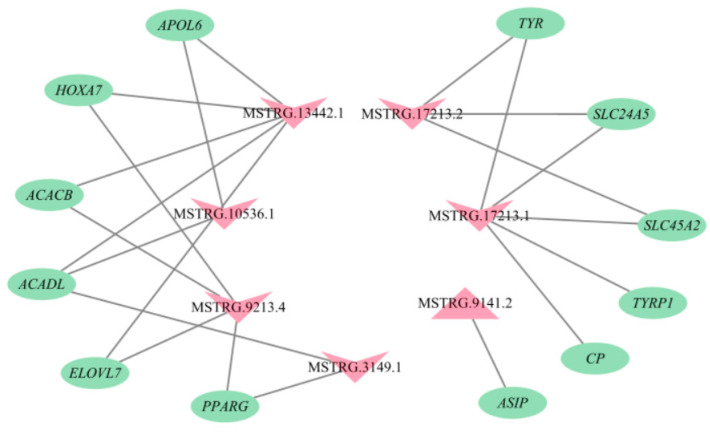
The lncRNA-mRNA interaction network. The pink triangles and inverted triangle exhibit up-regulated and down-regulated lncRNAs in skin tissue of Liaoning cashmere (LC) goats compared to Ziwuling black (ZB) goats, respectively. The green ellipses show the target genes.

**Table 1 genes-14-00384-t001:** The name, sequences and amplicon size of RT-qPCR primers.

Name	Forward (5′→3′)	Reverse (5′→3′)	Amplicon Size (bp)
MSTRG.18612.3	GTTGAAAAGAACTTTGAAGAGAGAG	CGGAGGGAGGGCGGGTGGAGGG	180
MSTRG.13442.1	GTCCAGCTCTCTGCAACCCCGTGGA	CAGGAGATGTAGGCGACCC	151
MSTRG.12234.1	TAACTGGTTAGTCTAATGGTCTGTT	GGTTCCAGGTCAATCCAG	150
MSTRG.18609.1	CCAGTCAGTGTAGCGCGCGTGCAGC	CGATTCCGTGGGTGGTGGTGCA	231
MSTRG.18613.1	GGGGCCGGCGGGGGACCGCCCCCCG	CGCGGCGACGAGGGCTGGCTC	241
MSTRG.300.17	GCTGCCCTTTCCTGTAAGCA	GGTTGGGCAGAGAGACTCGT	186
MSTRG.2390.1	GGCCTCAGTAGACAGTTGACAGGGT	TGAAGGATGACAGTGGGAAG	245
MSTRG.15153.3	ATCACCTCCCGTTGTCTCTC	AGTTCAGCTTGGAGTGGGAC	129
MSTRG.12739.2	CAGGGTTGGTTGAGCAGGCAGCTGG	GGAGCAAGTGGGAATGGGTA	192
*GAPDH*	ATCTCGCTCCTGGAAGATG	TCGGAGTGAACGGATTCG	227

## Data Availability

The datasets presented in this study can be found in online repositories. The names of the repository/repositories and accession number(s) can be found below: NCBI [accession: SRR19879981-SRR19879992]. The original contributions presented in the study are included in the article/Appendix A.

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
