# Peer review of "RNA-Seq Reveals the Roles of Long Non-Coding RNAs (lncRNAs) in Cashmere Fiber Production Performance of Cashmere Goats in China"

_genes, 2023, doi:10.3390/genes14020384_

Round 1

Reviewer 1 Report

This is an interesting manuscript focused to study the expression profiles of lncRNAs in skin tissue from Liaoning cashmere and Ziwuling black goats with significant difference in cashmere yield, cashmere fiber diameter and cashmere color using RNA sequencing (RNA-seq). The methodology is described with sufficient detail. Results are clear but some information appears to be missing. Discussion is well supported. References writing should be revised.

I suggest considering next minor comments to improve the manuscript:

-       Line 34: The references numbers should be in square brackets. Please correct throughout the manuscript.

-       Line 36: Remove the word “it”.

-       Lines 46-47: This sentence is a little confusing; please check it.

-       Line 57: Remove the words “Tibetan cashmere goats”.

-       Line 61: Replace the comma by a semicolon. Remove the word “that”.

-       Line 122: Replace “normalize” by “normalized”.

-       Line 136: Table 1 is missing.

-       Line 255: Replace “Of” by “Of the”.

-       Lines 255-258: I suggest to briefly describing main results from the network.

-       Line 284: Replace “On contrary” by “Conversely”.

-       Line 296: Replace “that” by “and”.

-       Lines 306-315: This sentence is very long and a little confusing. Please separate it in 2 or 3 short and clear sentences.

-       Line 333: Insert the word “and” after the comma.

-       Line 359: Replace “lncRNA” by “lncRNAs”.

-       Line 383: In References section, all references should be corrected following the guidelines described in the “Instructions for Authors”. Also, in several references the list of authors is incomplete.

Author Response

Line 34: The references numbers should be in square brackets. Please correct throughout the manuscript.

AU: The reference numbers of the manuscript have been placed in square brackets.

Line 36: Remove the word “it”.

AU: We have revised this in line 37.

Lines 46-47: This sentence is a little confusing; please check it.

AU: We have checked and revised this in lines 51-52.

Line 57: Remove the words “Tibetan cashmere goats”.

AU: We have revised this in line 63.

Line 61: Replace the comma by a semicolon. Remove the word “that”.

AU: We have revised this in line 66.

Line 122: Replace “normalize” by “normalized”.

AU: We have revised this in line 136.

Line 136: Table 1 is missing.

AU: We add Table 1 in line 151.

Line 255: Replace “Of” by “Of the”.

AU: We have revised this in line 277.

Lines 255-258: I suggest to briefly describing main results from the network.

AU: We have described main results from the network in lines 280-286.

Line 284: Replace “On contrary” by “Conversely”.

AU: We have revised this in line 317.

Line 296: Replace “that” by “and”.

AU: We have revised this in line 327.

Lines 306-315: This sentence is very long and a little confusing. Please separate it in 2 or 3 short and clear sentences.

AU: We have separated the sentences in lines 337-348.

Line 333: Insert the word “and” after the comma.

AU: We have revised this in line 369.

Line 359: Replace “lncRNA” by “lncRNAs”.

AU: We have revised this in line 404.

Line 383: In References section, all references should be corrected following the guidelines described in the “Instructions for Authors”. Also, in several references the list of authors is incomplete.

AU: All references have been corrected according to the guidelines described in "Instructions for Authors".

Reviewer 2 Report

Manuscript provides limited information to readers. Authors should explain why two different of breed used in the experimantal design. How authors can explain expression difference in a such experimental design. 

Author Response

Manuscript provides limited information to readers. Authors should explain why two different of breed used in the experimantal design. How authors can explain expression difference in a such experimental design.

AU: We have added the information of Liaoning cashmere goats and Ziwuling black goats in the Introduction section in lines 70-77.

Liaoning cashmere (LC) goats and Ziwuling black (ZB) goats are the main cashmere goat breeds in China and of economic importance for goat farmers. The two breeds have distinct different cashmere fiber traits. LC goats produce pure white cashmere fiber with higher fiber yield and fiber diameter. In contrast, ZB goats produce purple cashmere fiber with lower fiber yield and fiber diameter. Specifically, the average cashmere fiber yield and cashmere fiber diameter of LC goat are 1300 g and 15.5 μm, respectively, while the average cashmere yield and cashmere fiber diameter of ZB goats are 310 g and 14.1 μm, respectively. However, the molecular mechanism that regulates the differences in cashmere fiber traits between LC goats and ZB goats remains unclear. Accordingly, the experiment was designed to uncover the molecular mechanism underlying the difference. Because goats from the two breeds were raised at the same feeding and management levels in the study and they had also same gender and age, the expression difference of lncRNAs may reflect the difference in cashmere fiber color, cashmere fiber yield and cashmere fiber diameter between LC goats and ZB goats. The contribution of differentially expression lncRNAs to the differences of the traits have been fully discussed in the Discussion section.

Reviewer 3 Report

In this study, the authors wanted to investigate the role of lncRNA in determining cashmere fiber characteristics in goats. To this end, they compared skin tissue transcriptomics data from six Liaoning cashmere (LC) goats and six Ziwuling black (ZB) which differ in cashmere fiber yield, diameter, and color.

They performed lncRNA analysis, obtaining quantification and differential expression results for both analyses. Moreover, they infer a lncRNA-mRNA network to determine the most enriched signal pathways in which they are involved.

The objective of this study is quite interesting, and it has an appropriate experimental design, nevertheless, I would like to have elucidation regarding some points:

1. Can you please better explain based on which rules are you defining cis and trans lncRNA functions? I mean if I well understood, you are considering cis regulations for the closest genes (100Kb), while for a trans regulation, you are considering the Pearson correlation.

Are you considering positive, negative, or both kinds of correlations? Indeed, lncRNA can regulate gene expression in a different way and turn increase or decrease gene expression.

In my opinion, it would be good to calculate the Pearson correlation for both cis and trans target genes, you can include a heatmap and a table with the r values, which could also help to better understand the lncRNA mechanism involved in regulating the different gene targets.

2. In the introduction I would include more details regarding the importance and mechanism of lncRNA.

3. Line 112: include the meaning of the selected class code “u, i, j, x, c, e, o” will be better.

4. Line 116: Can you include a Venn diagram of intersected and unique results from two software used?

5. Line 36: in the text, you are referring to goats while the references (2, 3) are regarding sheep

6. Line 46-47: can you reword this sentence, when you refer to goat and/or sheep it is confusing

7. Line 139: “differentially expressed genes screened previously”, do you mean in a previous study? Is that published? If not, you need to insert the pipeline description of gene differential expression analysis in another paragraph of the material and methods section.

8. Supplementary materials are not included I could not review them. Please, can you include them in your answer?

Author Response

1. Can you please better explain based on which rules are you defining cis and trans lncRNA functions? I mean if I well understood, you are considering cis regulations for the closest genes (100Kb), while for a trans regulation, you are considering the Pearson correlation.

Are you considering positive, negative, or both kinds of correlations? Indeed, lncRNA can regulate gene expression in a different way and turn increase or decrease gene expression.

In my opinion, it would be good to calculate the Pearson correlation for both cis and trans target genes, you can include a heatmap and a table with the r values, which could also help to better understand the lncRNA mechanism involved in regulating the different gene targets.

AU: The basic principle of cis target gene prediction holds that the function of lncRNA is related to its neighboring protein coding genes, and the upstream lncRNA may intersect with promoters or other cis-acting elements of co-expressed genes, thus regulating gene expression at transcription or post-transcription level. LncRNA located downstream of 3'UTR or gene may be involved in other regulatory actions. Therefore, we annotate the lncRNA which was annotated as "unknown region" in the analysis. If it is located within 100kb of the upstream or downstream of a gene, these lncRNA may intersect with the region where cis-acting elements are located, thus participating in the process of transcription regulation. The basic principle of trans target gene prediction holds that the function of lncRNA has nothing to do with the position of coding gene, but is related to the protein coding gene co-expressed with it. The target gene can be predicted by correlation analysis or co-expression analysis between lncRNA and protein coding gene between samples.

Because the analysis of expression correlation needs a certain number of samples to ensure the accuracy of its prediction, when the number of samples is less than 6, the analysis is not carried out. When the number of samples is ≥ 6, Pearson correlation coefficient method is used to analyze the expression correlation between lncRNAs and protein coding genes, and the protein coding genes with absolute correlation greater than 0.95 are selected for GO/Pathway function enrichment analysis to predict the main functions of lncRNAs.

The expression of lncRNA is positively or negatively correlated with its trans target gene. Because the regulation of lncRNAs on the target gene in cis is related with the location of the target gene (within 100 kb), but not its expression, the correlation in expression between lncRNA and its cis target gene could not be calculated.

Pearson correlation coefficient in expression between differentially expressed lncRNAs with its trans target gene was calculated. Only the genes with |r| > 0.95 and P-value < 0.05 were selected as trans target genes of differentially expressed lncRNAs. These have been described in lines 157-161. The Pearson correlation in expression between lncRNA and its target gene can be found in Supplementary File 3.

2. In the introduction I would include more details regarding the importance and mechanism of lncRNA.

AU: We have added the importance and mechanism of lncRNAs in lines 46-50.

3. Line 112: include the meaning of the selected class code “u, i, j, x, c, e, o” will be better.

AU: We included the meaning of the selected class code “u, i, j, x, c, e, o” in lines 123-127.

4. Line 116: Can you include a Venn diagram of intersected and unique results from two software used?

AU: We added the Venn diagram in figure 1A.

5. Line 36: in the text, you are referring to goats while the references (2, 3) are regarding sheep.

AU: We have revised the references (2, 3) in lines 435-444.

6. Line 46-47: can you reword this sentence, when you refer to goat and/or sheep it is confusing.

AU: We have revised this in line 52.

7. Line 139: “differentially expressed genes screened previously”, do you mean in a previous study? Is that published? If not, you need to insert the pipeline description of gene differential expression analysis in another paragraph of the material and methods section.

AU: We have revised this in line 155.

8. Supplementary materials are not included I could not review them. Please, can you include them in your answer?

AU: We have loaded supplementary materials. The reviewer can also found them by clicking on the following link.

https://www.jianguoyun.com/p/Dfgebs8Qlc29Chjv1vMEIAA.

Reviewer 4 Report

This manuscript has a solid methodology from technical point of view, also it is well-written. There ares some weaknesses:

-the topic is far from application in science.

-The lncRNA interaction with miRNA is not considered. It could be done easily before publication.

Author Response

The topic is far from application in science.

AU: We have added the description about application prospect in lines 404-406. We screened out the key lncRNAs that may regulate cashmere traits by RNA-seq combined with bioinformatics analysis in the study. It needs further study the functional and molecular mechanisms of these lncRNA in the regulation of fiber follicle development and cashmere traits, eventually providing a theoretical basis for the application of cashmere fiber traits improvement.

The lncRNA interaction with miRNA is not considered. It could be done easily before publication.

AU: We have added lncRNA interaction with miRNA in the Materials and Methods section in lines 171-174, the Result section in lines 291-295 and the Discussion section in lines 390-398.